# Inducing Document Representations with Graph-Based Methods: A Blueprint

**Boshko Koloski**[1,2*] **Marko Pranjić** [1,2] **Nada Lavrač** [1,2] **Blaž Škrlj**[1*], **Senja Pollak** [1]
\* equal contribution
[1] Jozef Stefan Institute, Ljubljana, Slovenia, Ljubljana, Slovenia
[2] Jozef Stefan International Postgraduate School, Ljubljana, Slovenia
[3] School of Engineering and Management, University of Nova Gorica, Vipava, Slovenia
{name.lastname@ijs.si}

## Abstract

Representing documents in continuous numerical spaces is one of the key tasks in NLP. Contemporary state-of-the-art techniques leverage large neural networks and learn the document representations self-supervised. However, while these approaches excel at learning contextual word representations, they often overlook implicit document-to-document relations that can arise in real-world settings. We propose a blueprint method for constructing document representations that explicitly accounts for such implicit relations to address this issue.

## 1 Introduction

In contemporary NLP, researchers usually model documents as sequential collections of words using techniques such as (recurrent) neural networks Mikolov et al. (2013b;a); Peters et al. (2018)) or attention-based transformer NNs (Devlin et al. (2018); Liu et al. (2019)) to capture contextual relationships between words. Alternative methods for constructing document representations involve building graphs from co-occurring words and using them to represent the documents (Bunke & Riesen (2011); Sonawane & Kulkarni (2014); Yao et al. (2019); Osman & Barukub (2020)). These graph-based document representations have shown promise in capturing not only the sequential relationships between words but also the global structure of the document, performing on par with language model approaches (Zhang et al. (2020); Ragesh et al. (2021); Wang et al. (2023)), allowing for more effective and interpretable representations that can be useful in various downstream NLP tasks (like document similarity (Paul et al. (2016)), topic modeling (Xie et al. (2021)) and document understanding (Gemelli et al. (2023))). Huang et al. (2022) presented a method that enhances the language model's representation by incorporating information from a sub-word co-occurrence graph using a shared loss function. For tasks that involve analyzing documents within a network structure, such as identifying fake news (Han et al. (2020)) or recommending items based on textual reviews Fan et al. (2019), Graph Neural Networks (GNNs) have proven to be highly effective (Wu et al. (2023)). However, real-world documents often lack explicit connections between them, making it difficult to apply GNNs. We present a framework for constructing a document-to-document (D2D) network (available here https://github.com/bkolosk1/doc2doc.git) and evaluate various GNNs.

## 2 Graphs of documents and where to find them?

Let $G = (V, E)$ be a graph, where $V$ is the set of nodes (in our case, the set of documents) and $E$ is the set of edges. In our case the set of edges $E$ is not given and the goal is to construct it with the examination of the potential of documents to form links. The first step of our method is to transform the documents from raw texts to $d$-dimensional continuous vector space $L \in \mathbb{R}^d$. Next, we calculate the the edge weight as follows:

$$e_{i,j} = sim(L(v_i), L(v_j)) \text{ where } L \text{ denotes the latent document embedding}$$

Here, $v_i$ denotes a document, and $L(v_i)$ the latent embeddings of that document. The function $sim$ measures similarity between two embeddings, representing the edge weight $e_{i,j} \in \{0, 1\}$ between $v_i$

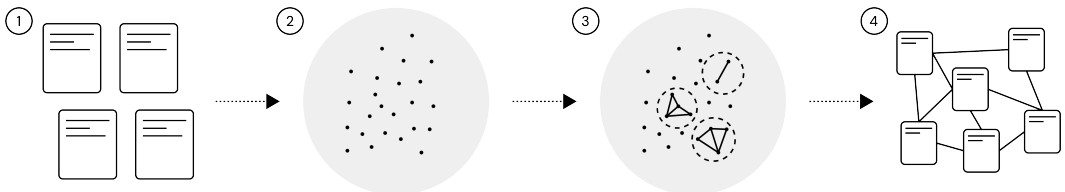

Figure 1: The proposed methodology aims to robustly create a network of related documents.

and $v_j$. Finally, we apply the thresholding function $thresh$ and keep only the edges that the function discriminates. We examine a transductive and an inductive scenario for obtaining a representation of a new document. The approach is presented in Figure 1 and the algorithm in Appendix 1.

**Everything and all at once.** In this scenario, we construct a graph between all the documents in the corpus. The graph is built by first computing their latent embeddings and later calculating their similarities. We hypothesize that a given document will have strong connections only with a small subset of the documents. To address this, we relax the constraints and compute the closest $k$ documents for each document in $D$. We normalize the documents and construct a KDTree Qiu et al. (2018) of distances, wherethe closest $k$ documents are the most similar in terms of cosine distance.

**The whole is the same as the sum of its parts.** In this scenario, we attempt to induce representations for new, previously unseen documents using the existing graph structure. We begin by constructing the graph using the aforementioned approach only from the training set of documents. Next, we use the NetMF embedding to obtain node (document) representations. To induce a representation for a new document, we first query the KDTree to retrieve the $k$ closest neighbors of the paper. We build the final document representation by averaging over the retrieved neighbour node embeddings.

## 3 EXPERIMENTAL SETUP AND EVALUATION

To evaluate our method, we choose six different text classification datasets. We benchmark our method against two SOTA language models, LinkBERT (Yasunaga et al. (2022)) and RoBERTa (Liu et al. (2019)), both fine-tuned for 10 $epochs$ with default parameters. We use MPNet as the latent space representation (Reimers & Gurevych (2019)). We explore a grid of hyper-parameters for our method with $thresh \in \{0.1, 0.9\}$, and $k \in \{5, 100\}$. For the transductive models, we use the Spectral implementation of GCN (Kipf & Welling (2016)) and GAT (Veličković et al. (2017))) with default parameters. For NetMF (Qiu et al. (2018)), we use the implementation by Škrlj et al. (2020) and train a LogisticRegression classifier with C=1. Table 1 summarizes the evaluation results (we select the models that performed the best on the dev. set with respect to the F1-Macro.

Table 1: Dataset information and averaged test data evaluation on the 10 runs with different seeds.

| | Dataset description | | | | Language Models | | Transductive D2D | | Inductive D2D |
|---|---|---|---|---|---|---|---|---|---|
| Dataset | Train Size | Dev Size | Test Size | Classes | LinkBERT | RoBERTa | GAT | GCN | NetMF3 |
| BBC (Greene & Cunningham (2006)) | 790 | 264 | 352 | 4 | **0.9863** | 0.9815 | 0.6404 | 0.7937 | 0.9460 |
| MBTI (Myers (1962)) | 4879 | 1627 | 2169 | 16 | **0.5695** | 0.3468 | 0.5072 | 0.1341 | 0.2116 |
| AAAI-FN (Patwa et al. (2021)) | 6420 | 2140 | 2140 | 2 | 0.9802 | **0.9819** | 0.8229 | 0.7380 | 0.9120 |
| PAN-Age (Rangel et al. (2016)) | 225 | 76 | 101 | 5 | 0.3881 | 0.4356 | **0.7335** | 0.3445 | 0.4554 |
| PAN-Gender (Rangel et al. (2017)) | 2024 | 675 | 900 | 4 | **0.7523** | 0.7444 | 0.6823 | 0.5669 | 0.6078 |
| PAN-FakeNews (Rangel et al. (2020)) | 270 | 30 | 200 | 2 | 0.6475 | **0.6725** | 0.6697 | 0.5629 | 0.6450 |

## 4 CONCLUSION AND FURTHER WORK

Our method performs comparably with some SOTA techniques, especially with numerous classes and small samples. When more data is available, the method's performance falls compared to the LLMs. To improve our method, we plan to improve the thresholding step, add more layers to the network (either via different embedding methods or via metadata), and speed-up the similarity search with fuzzy-search (e.g. FAISS). We believe the induced graph structure can transform NLP tasks into graph-theoretic ones and vice versa (like clustering with community detection).

## ACKNOWLEDGMENTS

We acknowledge financial support from several sources: the Slovenian Research Agency via research core funding for the programme Knowledge Technologies (P2-0103) and the projects CAN-DAS (Computer-assisted multilingual news discourse analysis with contextual embeddings, J6-2581) and SOVRAG (Hate speech in contemporary conceptualizations of nationalism, racism, gender and migration, J5-3102). A Young Researcher Grant PR-12394 supported the work of the first author. We thank Mihajlo Tunev for the improved scheme illustration.

## URM STATEMENT

All authors meet the URM criteria of ICLR 2023 Tiny Papers Track.

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

## APPENDIX 1: ALGORITHM

We present the pseudocode that outlines the steps involved in constructing the D2D network. By following these steps, we can establish a network that captures the relationships between different documents based on a $SimilarityFunction$, from a $LattentEmbedding$ space based on the $ThresholdingFunction$, and a $NearestNeighbourAlgorithm$ for search of the target $kNeighbors$. The algorithmic approach we use is detailed below.

---

**Algorithm 1** D2D network construction algorithm

---

**Require:** Documents, SimilarityFunction, LatentEmbedding, NearestNeighbourAlgorithm, ThresholdingFunction, kNeighbours

  documentNetwork ← Empty
  documentEmbeddings ← LatentEmbedding(Documents)
  nnSearch ← NearestNeighbourAlgorithm(documentEmbeddings)
  **for** document ∈ Documents **do**
    TargetDocuments ← nnSearch(document, kNeighbours)
    **for** targetDocument ∈ TargetDocuments **do**
      edgeWeight ← SimilarityFunction(document, targetDocument)
      **if** ThresholdingFunction(edgeWeight) **then**
        documentNetwork.makeEdge(document, targetDocument, edgeWeight)
      **end if**
    **end for**
  **end for**
  **return** documentNetwork

---

**Time complexity** In our approach, first, we use MPNet for *LatentEmbedding*, which infers the embedding of an article in constant time $c_{latent}$, in total $\mathcal{O}(D \cdot c_{latent}) = \mathcal{O}(D)$. To enable efficient search, we incorporate a KDTree as *NearestNeighbourAlgorithm*, which can be constructed in $\mathcal{O}(D \log D)$ time. The core component of our algorithm is the construction of the adjacency matrix which is based on the number of target neighbours $k$. The overall complexity of our approach in the worst-case scenario where $k = D$ becomes $\mathcal{O}(D^2)$.

