# OpenReview forum: "Inducing Document Representations from Graphs: A Blueprint"
_ICLR.cc/2023/TinyPapers — Submitted to Tiny Papers @ ICLR 2023_

### Official Review · Reviewer_6Nj4 · 2023-03-28

**Confidence:** 2

**Summary Of Contributions:**

This paper presents a framework for constructing a doocument-to-document network and demonstrate how various GNNs can be applied to this structure.

**Rating:**

High Potential (HP): a submission which meets the reviewing criteria and has potential to make an impact on the field

**Strengths And Weaknesses:**

This is a nice contribution in the field of document representation based on graph modeling. The authors show that in scenarios where there are numerous classes and small number of samples, their approach works on par with current state of the art techniques.
One scaling concern is that how practical is such an approach to large collections of documents.

**Suggested Changes:**

Since the there is a limit for this workshop (at most 2 pages of main text), I suggest making this work shorter and more concise. Moving some sections to an appendix would another option.

Typo: doocument -> document

---

### Meta-Review · Area_Chair_j5p8 · 2023-04-09

**Recommendation:** Invite to archive
**Confidence:** 3

**Metareview:**

The paper provides a novel framework for creating document representations from graphs of latent embeddings. It's an interesting approach to modeling document similarity that performs better in select circumstances, but is vastly outperformed by Transformer embedddings in most cases.

As noted in the other review, the scaling concern in terms of large document sets is one that should be addressed, either in terms of actual time required or time complexity analysis.

Additionally, it might be valuable to create a comparison to a simpler model like a byte-pair embedding to see what improvements this system can create over that before going straight to Transformer style architectures.





**Summary:**

The paper proposes a new method to compute document similarity from their graph representations. The reviewer states that the article performs at par for small number of samples per class, but might not scale to large document sets

**Comments And Feedback To The Authors:**

There are two typos saying "doocument"

If possible in the review, provide a time complexity analysis of your methods

**Reason For Not Giving A Higher Recommendation:**

### Clarity

- Are the findings communicated clearly and effectively? - **Yes**
- Does the paper include appropriate discussion of other relevant literature? - **Yes**
### Correctness

- Are the claims and conclusions justified by the findings? - **Yes**

### Reproducibility
- Does the paper describe its methods in such detail that a reader could independently reproduce the findings? - **No**
e.g., for empirical findings, are code and/or data provided?

Note: The paper is also longer than the limit without acknowledgements

**Reason For Not Giving A Lower Recommendation:**

N/A

---

### Decision · Program_Chairs · 2023-04-09

Invite to archive